

# The impact of a disaster medicine clinical training program on medical students' disaster literacy

Aysel Başer[1] and Zeynep Sofuoğlu[2]

[1] Department of Medical Education, Faculty of Medicine, Izmir Democracy University, İzmir, Turkey
[2] Department of Public Health, Faculty of Medicine, Izmir Democracy University Buca Seyfi Demirsoy Training and Research Hospital, İzmir, Turkey

Corresponding author
Aysel Başer, aysel.baser@idu.edu.tr

## ABSTRACT

**Background.** This study was conducted to assess how students' disaster literacy was affected by the Disaster Medicine Clinical Training Program at the Izmir Democracy University Faculty of Medicine (IDUFM) during the academic year 2022–2023.

**Methods.** Using an experimental method involving experimental and control groups, measurements were made at different times. The sample consisted of 5th-year students at IDUFM for the experimental group, while the control group was composed of 3rd- and 4th-year students from different buildings with limited interaction with the experimental group. The clinical training program was organized to improve the clinical practice skills of students in the field of disaster medicine and provide theoretical information. Throughout their internship, the students were provided with a program including information on types of disasters, preparation, response, relief, emergency surgical procedures, hospital disaster plans, national and international organizations and institutions working in disasters, and other relevant topics. The Disaster Literacy Scale (DLS) was used to collect data. This scale is a self-report scale that was developed to evaluate knowledge levels about disasters. In this study, the scale was applied as a pretest and a posttest, and the obtained data were analyzed using independent samples and paired-sample $t$-tests.

**Result.** The DLS scores of the experimental group showed significant improvement after the training program compared to their pretest scores (Total Scale: $p < 0.001$, Damage Mitigation: $p = 0.002$, Preparation: $p < 0.001$, Response: $p < 0.001$, and Relief: $p = 0.004$). When comparing the posttest results of the experimental group with the control group, the experimental group demonstrated significantly higher scores in Total Scale ($p = 0.01$), Damage Mitigation ($p = 0.02$), Response ($p = 0.03$), and Relief ($p < 0.001$). However, no significant differences were observed between the experimental group's pretest (T1) scores and the control group's posttest (T3) scores ($p > 0.05 p > 0.05 p > 0.05$), indicating that the knowledge levels of the experimental and control groups were homogeneous prior to the training intervention. These findings confirm the effectiveness of the Disaster Medicine Training Program in improving disaster literacy and response skills among medical students.

**Conclusion.** This study, which aimed to determine the effects of the Disaster Medicine Clinical Training Program on the disaster literacy of medical students, revealed that the program increased the literacy levels of the students and contributed to their responsible decision-making. It is thought that such education programs can make significant contributions to the effective management of healthcare services in disaster situations.

# INTRODUCTION

Disasters, unexpected and potentially devastating events, can manifest in various forms, including natural calamities, technological mishaps, disease outbreaks, and acts of terrorism. These events can profoundly impact public health, infrastructure, economy, and social structures (*Afzali & Viggers, 2015*; *Özüçelik, 2019*).

Insufficient precautions against disasters can lead to economic, environmental, social, physical, and mental challenges. While disaster management aims to mitigate these issues, sudden surges in physical and mental health problems, along with public health threats, can arise. This underscores the need for specialized healthcare professionals in disaster management. Studies following Hurricane Katrina highlighted a lack of basic disaster preparedness and response training among doctors, contributing to negative patient outcomes (*Hamm, 2006*; *Leder & Rivera, 2006*; *Scott, Carson & Greenwell, 2010*; *Afzali & Viggers, 2015*).

Disaster medicine encompasses preparation, planning, intervention, and response before, during, and after sudden and unexpected natural or anthropogenic disasters. It aims to effectively manage healthcare services and facilitate emergency medical interventions during disasters. Given the absence of an official disaster medicine program in Türkiye, post-graduation knowledge in disaster management becomes essential for all physicians (*TUKMOS, 2023*). Providing prompt and effective healthcare during disasters hinges on physicians having accurate information and appropriate skills. Hence, pre- and post-graduation training, such as disaster medicine internships, is of paramount importance (*Hamm, 2006*; *Leder & Rivera, 2006*; *Scott, Carson & Greenwell, 2010*).

In recent years, disaster literacy has gained significance in disaster science. It relates to individuals' preparedness to implement complex disaster response strategies in modern society. Disaster literacy involves understanding threats to individuals, families, and communities, developing attitudes toward factors affecting these threats, and assessing these factors. Its goal is to enhance societal resilience against disasters (*Muktaf, Ip & Ikom, 2017*; *Çalışkan & Üner, 2021*).

Since the 1980s, disaster medicine has advanced significantly, particularly post the September 11, 2001 terrorist attacks. Many countries have established disaster medicine programs (*Scott, Carson & Greenwell, 2010*; *European Society for Emergency Medicine, 2024*). However, Türkiye's progress in this field remains limited. The earthquakes experienced in Türkiye in February 2023 are expected to drive improvements. The Pre-Graduation Medical Education National Core Curriculum of Türkiye (NCC) outlines qualifications for knowledge, attitude, and skill development in Disasters and Extraordinary Circumstances (*Medical Deans Council (2020)*). An elective course was designed for 5th-year students at Izmir Democracy University Faculty of Medicine, incorporating a 2-week

Disaster Medicine Clinical Training Program. The program aimed to provide students with disaster information, along with UÇEP-defined qualifications. Physicians need the capacity to read, understand, and apply information. to enhance patient outcomes, make preparations, intervene, and contribute to relief efforts responsibly.

The study aimed to assess the effects of the 2022–2023 Disaster Medicine Clinical Training Program on the disaster information utilization skills of 5th-year students at Izmir Democracy University Faculty of Medicine (IDUFM), enabling responsible decision-making and instruction-following during disasters.

## MATERIALS & METHODS

This experimental study was conducted between March 2023-30 June 2023 at İzmir Demokrasi University Faculty of Medicine (IDUFM). The experimental design of the study included a pretest, a posttest, and a control group. The experimental and control groups consisted of undergraduate medical students from different classes of the same faculty.

Participants for the study were selected using a convenience sampling method. The experimental group consisted of 5th-year medical students enrolled in the Disaster Medicine Training Program at IDUFM. These students were chosen because they were scheduled to participate in the training as part of their curriculum.

The control group was composed of 3rd- and 4th-year medical students at IDUFM, selected due to their limited interaction with the experimental group. The control to experimental group ratio was approximately 65:60 = 1.1:1.

The total sample size was determined using G*Power analysis to ensure the study had sufficient power (80%) to detect a moderate effect size ($d = 0.5$) at a 5% significance level. Based on this analysis, a total of 128 participants (64 per group) was required. This was achieved through balanced recruitment from the experimental group population ($N = 67$) and the control group population ($N = 130$), resulting in 60 participants in the experimental group and 65 participants in the control group (Fig. 1).

### Experimental group

To investigate the effects of the training program on disaster literacy, the Disaster Literacy Scale (DLS) was applied to the experimental and control groups. For the pretest, the scale was administered first to the experimental group (5th-year medical students). After this, the training program was provided to the experimental group. The training was conducted over two weeks and included both theoretical and practical components designed to enhance their knowledge and skills in disaster medicine.

### Control group

To maintain the integrity of the control group, participants were 3rd- and 4th-year medical students at IDUFM who were housed in separate buildings with limited interaction with the experimental group. These students were not exposed to the training program during the study period, ensuring unbiased comparisons between groups. To further ensure objectivity, the knowledge and skill levels of students from different grade levels were compared, as the groups had no direct contact with each other. Given the recent
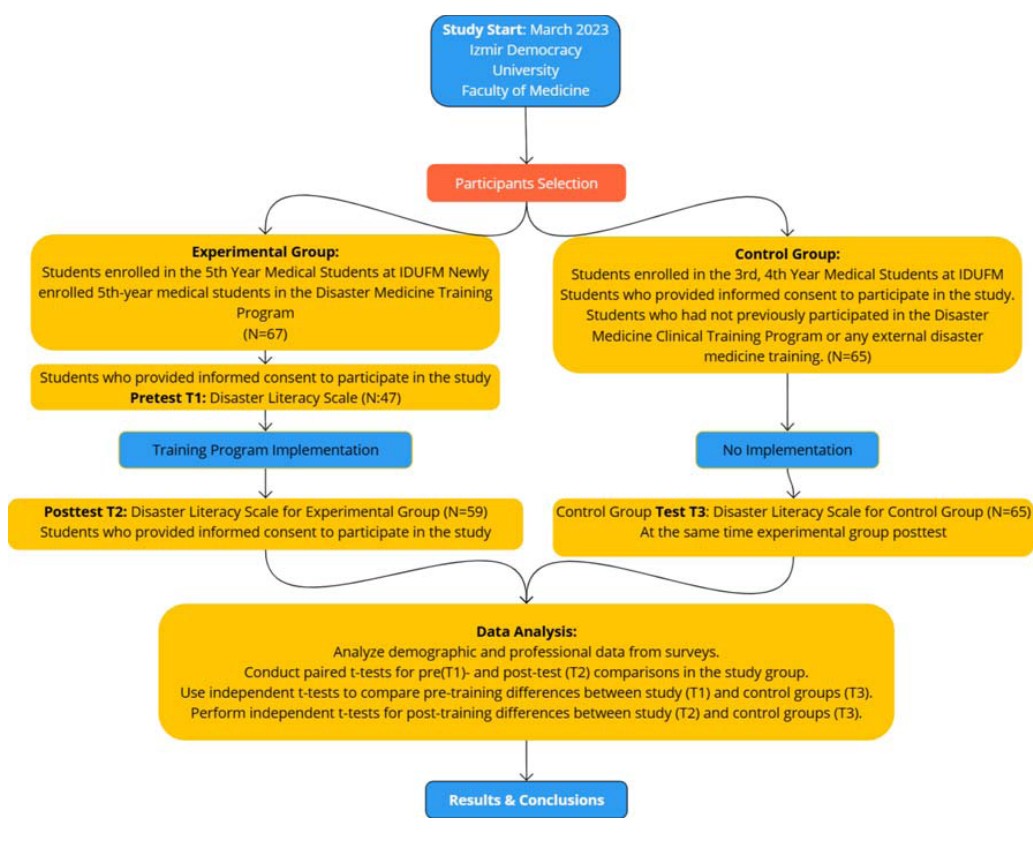

**Figure 1  Study design.**

disaster-related events in Türkiye, all students—regardless of their grade level—had likely been exposed to similar information through social media, family discussions, or personal research. As such, no significant baseline differences in disaster literacy levels between the 3rd, 4th, and 5th-year students were anticipated. To prevent external factors, such as media or general circumstances, from influencing the measurement results, the posttest was administered to the control group (3rd- and 4th-year students) at the same time it was applied to the experimental group, after the completion of the training program for the experimental group. This method ensured a robust comparison of the disaster literacy levels between the experimental and control groups. dditionally, as part of the curriculum, 3rd-year students will have the opportunity to voluntarily enroll in the disaster medicine training program after two years, and 4th-year students can opt to take it after one year. This ensures that the control group participants are not disadvantaged and have equal access to the training in the future.

To minimize potential biases arising from interactions between the experimental (5th-year students) and control groups (3rd- and 4th-year students), several measures were implemented. The groups were housed in separate buildings on the university campus and maintained distinct schedules, ensuring minimal physical interaction during the study period. The training program and related materials were exclusively accessible to the

experimental group, and no overlapping activities occurred. Additionally, the posttest for the control group was administered simultaneously with the experimental group's posttest, after the completion of the training program, to prevent any information leakage. These strategies ensured the integrity of the control group and the reliability of the study outcomes.

## Informed consent

Before the commencement of the educational program and data collection, all participants were thoroughly informed about the study's purpose and procedures. Verbal consent was obtained from the students in the study group at the beginning and end of the training program. Additionally, for the DLS scale administered online to both the control and study groups, participants were required to read and approve the informed consent form before proceeding with the scale.

This process ensured that participants voluntarily and informedly participated in the research. The study adhered to the principles outlined in the Declaration of Helsinki, and consent for publication was obtained from all participants.

## Training program

The Disaster Medicine Training Program at IDUFM is a two-week, interdisciplinary educational intervention designed to enhance disaster literacy and practical response skills among medical students. The program actively integrates contributions from a diverse range of professionals, involving faculty members and physicians from various specialties (such as pediatrics, public health, child and adolescent psychiatry, adult psychiatry, general surgery, orthopedics, obstetrics and gynecology, radiology, and medical education), nurses, paramedics, and emergency responders. Additionally firefighters and disaster management experts from organizations such as the Izmir Fire Department, AFAD and the Red Crescent.

## Program structure

This two-week program consists of theoretical and practical/applied sessions involving face-to-face teacher-student interactions and hands-on experiences. The program includes a total of 38 h of theoretical education and 20 h of practical sessions, culminating in 58 h of intensive training. Sessions are scheduled daily from 8:30 AM to 4:30 PM, with a blend of lectures, hands-on activities, and group-based learning.

Theoretical sessions: The theoretical component covers a broad spectrum of disaster-related topics:

Public health and preparedness: Disaster definitions, epidemiology, risk assessment, vulnerable groups, and infectious disease management.

Clinical management: Pediatric and neonatal care, trauma management (including crush injuries and compartment syndrome), obstetric emergencies, and early rehabilitation.

Specialized topics: Psychological first aid, psychosocial support, forensic evaluation, and trauma radiology.

Legislation and planning: National and international disaster frameworks, local disaster management plans (*e.g.*, TAMP), and public education in disaster scenarios.

Practical sessions: Practical activities emphasize hands-on experience and real-world applications, including:

Managing pediatric patients in disaster settings.

Conducting community education on disaster preparedness. Students will create posters and videos for community education on disaster preparedness.

Utilizing technological tools and resources in disaster management.

Visits to local emergency response facilities, including İzmir "Disaster and Emergency Management Authority" ("Afet ve Acil Durum Yónetimi Başkanlığı"-AFAD in Turkish), the Fire Department Education Center, and the Red Crescent.

Students gain practical skills in fire suppression, earthquake safety, and emergency response coordination during these visits.

Target disasters: The curriculum addresses a variety of disaster scenarios;

Natural disasters: Earthquakes, floods, and storms.

Human-induced disasters: Hazardous material incidents, fires, and mass casualty events.

## Data collection instruments

To measure the effects of the Disaster Medicine Clinical Training Program on disaster literacy, the 61-item DLS was used. DLS is a self-report scale that was developed by *Çalışkan & Üner (2023)* to evaluate the knowledge levels of individuals aged 18–60 about disasters. Permission to use the scale was obtained *via* email from Çalışkan on February 15, 2023. The approval confirmed the use of the scale for non-commercial research purposes, and its application through online survey platforms such as Google Forms or Microsoft Teams. This ensured the ethical and legal use of the instrument in our study.

DLS has a structure with four dimensions, namely, damage mitigation, preparation, response, and relief. It also involves information-gathering processes related to decision-making and practices about disasters (access, comprehension, decision-making, and implementation). Each item on the scale is scored as a 5-point Likert-type item, with scores varying from 1 to 5. There is no inversely scored item, and the total score range of the scale is 61 to 305. For ease of calculation, the total score of the respondent is standardized in the range of 0–50. DLS scores are divided into four categories, inadequate (<30), partially adequate (30–36), adequate (36–<42), and excellent (42–50), whereas its cut-off points are calculated based on an index [Formula=(arithmetic mean-1) × (50/4)]. Additionally, a categorization can also be made for each of the four dimensions of DLS (*Çalışkan & Üner, 2023*).

## Data analysis

In the study, a pretest was administered to the participants in the experimental group, while a posttest was administered to the participants in the experimental and control groups at the end.

In the analyses of the data, to test the normality of the distributions, mean, standard deviation, skewness, kurtosis, and Shapiro–Wilk test values were calculated. To determine whether there were significant differences between the DLS scores of the experimental and control groups because the data were distributed normally, $t$-tests, which are parametric

statistical tests, were conducted. In the comparisons, the level of statistical significance was accepted as $p < 0.05$. The SPSS 25.00 package program was used to analyze the data.

The skewness and kurtosis coefficients in different measurements in the experimental and control groups were between −1 and +1 (skewness: −0.19–0.51 and −0.50–0.18, respectively; kurtosis: −0.67–0.81 and −0.35–0.13, respectively) (*Çokluk, Şekercioğlu & Büyüköztürk, 2012*) and considering the Kolmogorov–Smirnov normality test results in addition to these values, the results of the control group test ($p = 0.855$) and those of the pretest of the experimental group ($p = 0.489$) were normally distributed. Therefore, both tests confirmed that the data were normally distributed (*Çokluk, Şekercioğlu & Büyüköztürk, 2012*). The reliability of the scales used in this study was measured using Cronbach's Alpha.

### Ethical approval

The data were collected from Izmir Democracy University Faculty of Medicine students after obtaining informed consent. The study was conducted in compliance with the ethical standards set forth by the Buca Seyfi Demirsoy Training and Research Hospital Non-Interventional Research Ethics Committee, affiliated with İzmir Democracy University (IDU). Ethical approval was secured prior to the initiation of the study (Ethics Committee Approval No. 2023/2- 133, Date: 22.02.2023).

## RESULTS

The study included 5th-year medical students who took part in the Disaster Medicine Clinical Training Program in the experimental group and 3rd- and 4th-year medical students who did not take part in the program in the control group. The population for the experimental group consisted of 65 students, while 60 students participated in the test. The number of students included in the control group was 65. One student in the experimental group was not included in the analyses because they did not complete the posttest. While 45% ($n = 27$) of the participants in the experimental group were female, 63% ($n = 41$) of the participants in the control group were female ($p < 0.001$). It was found that 6.7% ($n = 4$) of the participants in the experimental group and 29.2% ($n = 19$) of those in the control group had experienced a disaster before. The mean age of the participants in the experimental group was 24.10 (±4.10) ($p = 0.04$), and the mean age of those in the control group was 21.85 (±1.68) ($p < 0.001$) (Table 1).

The mean scale scores of the participants in the experimental group in the pretest (T1) were 32.26 (±5.76) for the damage mitigation dimension, 30.13 (±6.82) for the preparation dimension, 33.49 (±6.19) for the response dimension, 28.39 (±8.67) for the relief dimension, and 31.01 (±5.99) for the total scale. According to the score categories of the scale in the pretest, it was determined that the disaster literacy levels of the participants in the experimental group were inadequate in terms of the relief dimension and partially adequate in terms of the other dimensions and general DLS scores. The mean scale scores of the participants in the experimental group in the posttest (T2) were 35.27 (±7.73) for the damage mitigation dimension, 34.15 (±7.53) for the preparation dimension, 36.07 (±6.52) for the response dimension, 34.01 (±7.80) for the relief dimension, and 34.84 (±6.76) for the total scale. According to the score categories of the scale, after their participation

**Table 1** Sociodemographic characteristics of the participants.

| | | Experimental group | | Control group | |
|---|---|---|---|---|---|
| **Age** | | **Mean** | **Std. deviation** | **Mean** | **Std. deviation** |
| | | 24.10 | 4.11 | 21.85 | 1.69 |
| | | N | Percentage (%) | N | Percentage (%) |
| Gender | Female | 27.00 | 45.00 | 41.00 | 63.10 |
| | Male | 33.00 | 55.00 | 24.00 | 36.90 |
| | Total | 60.00 | 100.00 | 65.00 | 100.00 |
| Has experienced a disaster before | Yes | 4.00 | 6.70 | 19.00 | 29.20 |
| | No | 55.00 | 91.70 | 45.00 | 69.20 |
| | Do not want to answer | 1.00 | 1.70 | 1.00 | 1.50 |
| | Total | 60.00 | 100.00 | 65.00 | 100.00 |

in the training program, the disaster literacy levels of the participants in the experimental group increased from inadequate to partially adequate for the preparation dimension and from partially adequate to adequate for the response dimension, but although the mean scores of the participants increased in the other dimensions and the overall DLS, their levels remained partially adequate (Table 2).

The results, as presented in Table 3, show that there were no statistically significant differences between the experimental pre test (T1) and control groups (T3) in any dimension of the DLS and the total scale. The results for the experimental and control groups across each dimension and the total scale are as for Damage Mitigation, the experimental group scored 3.58 ($\pm$0.46) and the control group scored 3.57 ($\pm$0.61) ($p = 0.89$). In Preparation, the experimental group scored 3.41 ($\pm$0.55) and the control group scored 3.53 ($\pm$0.65) ($p = 0.30$). For Response, the experimental group scored 3.68 ($\pm$0.49) and the control group scored 3.66 ($\pm$0.63) ($p = 0.83$). In Relief, the experimental group scored 3.27 ($\pm$0.69) and the control group scored 3.35 ($\pm$0.77) ($p = 0.58$). Finally, for the Total Scale, the experimental group scored 3.48 ($\pm$0.48) and the control group scored 3.52 ($\pm$0.57) ($p = 0.68$). There was no statistically significant difference between the pretest (T1) results of the experimental group and the posttest (T3) results of the control group ($p > 0.05$). This finding strongly indicates that the experimental and control groups had similar literacy levels prior to the intervention (Table 3).

As seen in Table 4, the pretest (T1) and posttest (T2) DLS scores of the participants in the experimental group differed significantly from each other in all dimensions of DLS and the overall DLS (all values: $p = 0.00$). The DLS scores of the experimental group showed a general improvement, with a significant increase from the pretest to the posttest ($p < 0.001$), highlighting the overall positive impact of the training program on disaster literacy levels. These results showed that the disaster literacy levels of the participants in the experimental group increased after they took part in the training program, and they developed skills in this context (Table 4).

**Table 2  Descriptive statistics and skewness kurtosis values of the DLS total and subscale scores of the experimental and control groups for the scoring range of 0–50.**

| Group | | N | Minimum | Maximum | Mean | | Std. Deviation | Variance | Skewness | | Kurtosis | |
|---|---|---|---|---|---|---|---|---|---|---|---|---|
| | | Statistic | Statistic | Statistic | Statistic | Std. error | Statistic | Statistic | Statistic | Std. error | Statistic | Std. error |
| Experimental group | T1_DamageMitigation | 47 | 19.85 | 46.32 | 32.26 | 0.84 | 5.76 | 33.21 | 0.11 | 0.35 | −0.01 | 0.68 |
| | T2_DamageMitigation | 59 | 16.91 | 50.00 | 35.27 | 1.01 | 7.73 | 59.79 | −0.03 | 0.31 | −0.45 | 0.61 |
| | T1_Preparation | 47 | 14.16 | 50.00 | 30.13 | 0.99 | 6.82 | 46.48 | −0.07 | 0.35 | 0.81 | 0.68 |
| | T2_Preparation | 59 | 17.50 | 50.00 | 34.15 | 0.98 | 7.53 | 56.76 | −0.09 | 0.31 | −0.67 | 0.61 |
| | T1_Response | 47 | 18.27 | 50.00 | 33.49 | 0.90 | 6.19 | 38.28 | 0.16 | 0.35 | 0.70 | 0.68 |
| | T2_Response | 59 | 21.15 | 50.00 | 36.07 | 0.85 | 6.52 | 42.54 | −0.11 | 0.31 | −0.52 | 0.61 |
| | T1_Relief | 47 | 11.67 | 50.00 | 28.39 | 1.27 | 8.67 | 75.25 | 0.23 | 0.35 | −0.25 | 0.68 |
| | T2_Relief | 59 | 14.17 | 50.00 | 34.01 | 1.02 | 7.80 | 60.79 | −0.19 | 0.31 | −0.22 | 0.61 |
| | T1_ScaleTotal_Mean | 47 | 18.88 | 48.98 | 31.01 | 0.87 | 5.99 | 35.93 | 0.51 | 0.35 | 0.56 | 0.68 |
| | T2_ScaleTotal_Mean | 59 | 17.30 | 49.39 | 34.84 | 0.88 | 6.76 | 45.69 | −0.11 | 0.31 | −0.31 | 0.61 |
| Control group | T3_DamageMitigation | | 12.50 | 46.32 | 32.07 | 0.95 | 7.66 | 58.62 | −0.48 | 0.30 | −0.22 | 0.59 |
| | T3_Preparation | | 10.83 | 50.00 | 31.65 | 1.01 | 8.16 | 66.66 | −0.18 | 0.30 | −0.23 | 0.59 |
| | T3_Response | 65 | 14.42 | 48.08 | 33.20 | 0.98 | 7.87 | 61.98 | −0.50 | 0.30 | −0.35 | 0.59 |
| | T3_Relief | | 0.00 | 47.50 | 29.36 | 1.19 | 9.56 | 91.44 | −0.45 | 0.30 | 0.13 | 0.59 |
| | T3_ScaleTotal_Mean | | 13.91 | 45.67 | 31.53 | 0.89 | 7.17 | 51.43 | −0.35 | 0.30 | −0.24 | 0.59 |

**Table 3  DLS scores of experimental and control groups Pre-training program in experimental group.**

|  | Group | N | Mean | Std. deviation | Std. error mean | t | df | Sig. (2-tailed) |
|---|---|---|---|---|---|---|---|---|
| Damage Mitigation | Experimental group | 47.00 | 3.58 | 0.46 | 0.07 | 0.14 | 110.00 | 0.89 |
|  | Control group | 65.00 | 3.57 | 0.61 | 0.08 |  |  |  |
| Preparation | Experimental group | 47.00 | 3.41 | 0.55 | 0.08 | −1.04 | 110.00 | 0.30 |
|  | Control group | 65.00 | 3.53 | 0.65 | 0.08 |  |  |  |
| Response | Experimental group | 47.00 | 3.68 | 0.49 | 0.07 | 0.21 | 110.00 | 0.83 |
|  | Control group | 65.00 | 3.66 | 0.63 | 0.08 |  |  |  |
| Relief | Experimental group | 47.00 | 3.27 | 0.69 | 0.10 | −0.55 | 110.00 | 0.58 |
|  | Control group | 65.00 | 3.35 | 0.77 | 0.09 |  |  |  |
| Total Scale | Experimental group | 47.00 | 3.48 | 0.48 | 0.07 | −0.41 | 110.00 | 0.68 |
|  | Control group | 65.00 | 3.52 | 0.57 | 0.07 |  |  |  |

Table 5 presents the results of the independent samples $t$-test that was conducted to compare the DLS scores of the participants in the control group and the posttest DLS scores of the participants in the experimental group for testing the H3 hypothesis.

According to the results of the independent samples $t$-test, as seen in Table 4, there were statistically significant differences between the scores of the control group (T3) and the posttest scores of the experimental group (T2) in all dimensions and the total scale ($p < 0.05$), except for the preparation dimension ($p = 0.08$). The results showed that the disaster literacy levels of the participants increased with the program that was provided to them, and in this process, the measurement results were not influenced by external factors such as the media, any disaster situation, or the general circumstances of the country, except for the context of the preparation dimension. Accordingly, the training program was an important intervention that would create a difference in disaster literacy levels (Table 5).

The Cronbach's Alpha values for the experimental group in the pretest, posttest and control group are as follows: Pre-Test Experimental Group: Cronbach's Alpha = 0.944 with 61 items; Post-Test Experimental Group: Cronbach's Alpha = 0.968 with 61 items; Control Group: Cronbach's Alpha = 0.965 with 61 items. These values indicate that the scales used in the study have high reliability, suggesting that the data collected is consistent and dependable.

## DISCUSSION

It was completely coincidental that the Disaster Medicine Clinical Training Program that was integrated into the curriculum as a part of an elective course for the 2022–2023 academic year was completed before the earthquakes in Türkiye on 26 February 2023. After these earthquakes, it was decided to offer this program to all medical students.

There are very few studies on the design, feasibility, and effectiveness of disaster-related training for medical students (*Pfenninger et al., 2010*; *Scott, Carson & Greenwell, 2010*; *Scott et al., 2013*; *Afzali & Viggers, 2015*). Providing comprehensive disaster training for medical

Başer and Sofuoğlu (2025), *PeerJ*, DOI 10.7717/peerj.18800

**Table 4 Results of the paired-samples *t*-test for the pre-test and post-test DLS total and subscale scores of the experimental group for the scoring range of 0–50.**

| | *Paired samples statistics* | | | | Mean differences | Std. deviation | Std. error mean | 95% confidence interval of the difference | | t | df | Sig. (2-tailed) | DLS score category[*] |
|---|---|---|---|---|---|---|---|---|---|---|---|---|---|
| | Experimental groups pre- and posttest dimensions | N | Mean | Std. Deviation | | | | Lower | Upper | | | | |
| Pair 1 | Pre_Damage Mitigation (T1) | 47 | 32.45 | 5.68 | −3.88 | 8.02 | 1.18 | −6.27 | −1.50 | −3.28 | 45.00 | 0.002 | PA |
| | Post_Damage Mitigation (T2) | 47 | 36.33 | 7.57 | | | | | | | | | A |
| Pair 2 | Pre_Preparation (T1) | 47 | 30.08 | 6.88 | −5.10 | 8.45 | 1.25 | −7.61 | −2.59 | −4.09 | 45.00 | 0.000 | PA |
| | Post_Preparation (T2) | 47 | 35.18 | 6.66 | | | | | | | | | PA |
| Pair 3 | Pre_Response (T1) | 47 | 33.53 | 6.25 | −3.49 | 7.83 | 1.15 | −5.82 | −1.17 | −3.02 | 45.00 | 0.004 | PA |
| | Post_Response (T2) | 47 | 37.02 | 6.27 | | | | | | | | | A |
| Pair 4 | Pre_Relief (T1) | 47 | 28.32 | 8.76 | −7.07 | 8.54 | 1.26 | −9.60 | −4.53 | −5.61 | 45.00 | 0.000 | IA |
| | Post_Relief (T2) | 47 | 35.38 | 7.22 | | | | | | | | | PA |
| Pair 5 | Post_Scale Total (T1) | 47 | 31.04 | 6.06 | −4.90 | 6.97 | 1.03 | −6.97 | −2.83 | −4.77 | 45.00 | 0.000 | PA |
| | Post_Scale Total (T2) | 47 | 35.94 | 6.27 | | | | | | | | | PA |

**Notes.**

*DLS Score Category: inadequate (<30), partially adequate (30–36), adequate (36-<42), and excellent (42–50).

**Table 5  DLS scores of experimental and control groups after training program in experimental group.**

| | Group | N | Mean | Std. deviation | Std. error mean | Sig. | t | df | Sig. (2-tailed) |
|---|---|---|---|---|---|---|---|---|---|
| Post_Damage Mitigation | Experimental group | 59 | 35.27 | 7.73 | 1.01 | 0.88 | 2.31 | 122.00 | 0.02 |
| | Control group | 65 | 32.07 | 7.66 | 0.95 | | | | |
| Post_Preparation | Experimental group | 59 | 34.15 | 7.53 | 0.98 | 0.52 | 1.77 | 122.00 | 0.08 |
| | Control group | 65 | 31.65 | 8.16 | 1.01 | | | | |
| Post_Response | Experimental group | 59 | 36.07 | 6.52 | 0.85 | 0.15 | 2.20 | 122.00 | 0.03 |
| | Control group | 65 | 33.20 | 7.87 | 0.98 | | | | |
| Post_Relief | Experimental group | 59 | 34.01 | 7.80 | 1.02 | 0.09 | 2.95 | 122.00 | 0.00 |
| | Control group | 65 | 29.36 | 9.56 | 1.19 | | | | |
| Post_Total Scale | Experimental group | 59 | 34.84 | 6.76 | 0.88 | 0.64 | 2.63 | 122.00 | 0.01 |
| | Control group | 65 | 31.53 | 7.17 | 0.89 | | | | |

students is important for the success of future operations in the scope of disaster and emergency preparedness; however, very few medical schools have defined and implemented basic disaster medicine qualifications for healthcare professionals (*Scott et al., 2013*; *Kozyel et al., 2018*).

There are several studies on the exposure of students to various disasters (*Şen & Ersoy, 2017*; *Şahin, Lamba & Öztop, 2018*; *Yiğit et al., 2020*). For example, *Yiğit et al. (2020)* reported that 52.85% of medical and engineering students in Türkiye had experienced a disaster, with earthquakes being the most common at 46.20% (*Yiğit et al., 2020*). *Şahin, Lamba & Öztop (2018)* highlighted the prevalence of disasters in regions like Burdur, which experienced 122 earthquakes of magnitude 4.0 or greater between 1900 and 2015. Despite the high awareness levels among students, their disaster preparedness remained low (*Şen & Ersoy, 2017*; *Şahin, Lamba & Öztop, 2018*) emphasized significant gaps in disaster preparedness among hospital disaster teams in İzmir, a region with high earthquake risk, despite historical exposure to events such as the 1999 Marmara earthquake (*Şen & Ersoy, 2017*).

In the present study, only 6.7% ($n = 4$) of participants in the experimental group and 29.2% ($n = 19$) in the control group reported having experienced disasters. These rates are notably lower than those observed in previous studies. This discrepancy may be attributed to regional differences in disaster exposure or reporting. Additionally, the participants in this study were attending a university that enrolls students from across Türkiye. Many students may not have experienced disasters in İzmir or its surrounding areas, as they might originate from regions with lower disaster risk.

The disparity in disaster exposure rates underscores the importance of disaster education programs tailored to address varying levels of experience and preparedness. As *Şahin, Lamba & Öztop (2018)* noted, while disaster awareness is often high, actionable preparedness skills are frequently lacking (*Şahin, Lamba & Öztop, 2018*). Moreover, *Yiğit et al. (2020)* demonstrated that students who received disaster-related training had significantly higher preparedness scores (110.97 ± 12.86) compared to those without training (106.97 ± 12.43,

$p = 0.006$), further supporting the effectiveness of structured educational programs (*Yiğit et al., 2020*).

In the present study contributes to the growing body of evidence that comprehensive disaster education programs, combining theoretical knowledge with practical applications, can significantly enhance preparedness and response capabilities. By addressing both awareness and practical skills, such programs bridge critical gaps in disaster readiness, regardless of prior exposure to disasters.

The literature review that was performed for this study revealed that many studies have examined the disaster preparedness and knowledge levels of middle school students, high school students, the general public, and healthcare professionals in workplace settings (*Hamm, 2006*; *Leder & Rivera, 2006*; *Martin, Bush & Lynch, 2006*; *Şen & Ersoy, 2017*; *Tas, Cakir & Kadioglu, 2020*; *Labrague et al., 2021*; *Ranse et al., 2022*), while no study investigating the disaster literacy levels of medical students , in particular, was encountered. Other studies have examined the disaster preparedness and disaster awareness levels of university (*Şahin, Lamba & Öztop, 2018*; *Yiğit et al., 2020*). *Yiğit et al. (2020)*; the disaster preparedness levels of students were generally found to be low. A total of 78.8% of the students stated that they did not consider themselves prepared for potential disasters, and 94.3% reported not having a disaster kit (*Yiğit et al., 2020*). *Şahin, Lamba & Öztop (2018)*; while the disaster awareness levels of students were found to be high, their preparedness levels were reported as low. Additionally, 78.3% of the students stated that they did not feel prepared for disasters (*Şahin, Lamba & Öztop, 2018*). These findings indicate that although disaster awareness is relatively high, there are significant gaps in preparedness, highlighting the critical need for regular training and drills (*Şahin, Lamba & Öztop, 2018*; *Yiğit et al., 2020*). In this study, the DLS scores of the control group and the pretest-posttest scores of the experimental group were the lowest in the context of the preparedness and relief dimensions of the scale. A comparison could not be made as other studies did not include a relief dimension as a variable (*Şahin, Lamba & Öztop, 2018*).

*Afzali & Viggers (2015)*'s study highlights a 3-day disaster medicine course offered to medical students during the Anaesthesia and Reanimation Conference in May 2013 in Copenhagen, Denmark. According to the results of the mass injury simulation course, the students were interested in the topic of disaster medicine, and they could play an active and important role in the organization of such a course and planning of participation. *Afzali & Viggers (2015)* highlighted that structured educational programs, particularly those involving simulations and hands-on training, are essential for equipping medical students with disaster management skills and ensuring effective learning outcomes. They emphasized the need for integrating such courses into the medical school curriculum as part of an organized and continuous educational framework (*Afzali & Viggers, 2015*). When they graduate, medical students have limited time. During their undergraduate education, a significant part of their time is allocated to studying. It is important for this process to involve a structured program for disaster medicine (*Afzali & Viggers, 2015*). In medical schools, courses and internships are not organized on a semester basis but are rather determined as year-long or duration-based internships within an integrated education model. This integrated approach ensures that students acquire various clinical

and theoretical knowledge and skills over a longer and continuous period. Therefore, even though our training program was conducted in a short period of two weeks, it had a significant impact on the students. Additionally, no educational interventions were conducted for the control group, and careful measures were taken to minimize the impact of changes in the literature and recent disasters. In addition to findings suggesting the importance of integrating these programs into the curriculum based on the interest of students, our results demonstrated that the disaster literacy levels of the participants in the experimental group increased significantly as a result of the training program on disaster medicine. These results showed that the program that was organized was effective and played a significant role in improving disaster literacy skills. As suggested in the literature and emphasized by *Afzali & Viggers (2015)*, the Disaster Medicine elective internship program has been integrated into the curriculum of İzmir Democracy University Faculty of Medicine, reflecting a structured and sustainable approach to disaster education.

*Ranse et al. (2022)*'s study aimed to identify the educational needs of nursing students regarding disaster preparedness and to determine the priority content for inclusion in undergraduate nursing curricula. The study was conducted with clinical and academic nurses in Australia using a three-round Delphi method. The study identified high-priority statements such as "disaster knowledge," "assessment and triage," "critical thinking," and "technical skills." Additionally, statements related to "mental well-being" and "teamwork in stressful situations" were ranked highest in priority. The study concluded that disaster-related content should be included in the undergraduate nursing curriculum, either integrated into existing course units or offered as a standalone course (*Ranse et al., 2022*). Moreover, *Kınık & Çalışkan (2024)* emphasize the need for systematic educational interventions such as disaster literacy courses, which integrate teaching and health literacy principles to improve community resilience and individual decision-making in disaster contexts. Specifically, disaster literacy as described by *Kınık & Çalışkan (2024)* involves not only acquiring knowledge but also developing critical thinking, application, and evaluation skills to effectively mitigate risks and respond to disasters (*Kınık & Çalışkan, 2024*). This aligns with the goals of the present study, as both approaches underscore the importance of structured disaster education programs in enhancing preparedness and response capabilities among students. This aligns with the goals of the present study, as both approaches underscore the importance of structured disaster education programs in enhancing preparedness and response capabilities among students (*Ranse et al., 2022*; *Kınık & Çalışkan, 2024*). The findings of our study address key gaps highlighted in the literature, including the need for structured and comprehensive disaster education programs tailored to healthcare students. By combining theoretical knowledge with practical applications, our Disaster Medicine Clinical Training Program aligns with the systematic educational interventions proposed by *Kınık & Çalışkan (2024)* and the priorities identified by *Ranse et al. (2022)*. Furthermore, this study provides a unique contribution by implementing a disaster literacy-focused curriculum specifically for medical students, bridging the gap in existing disaster preparedness education and setting a precedent for future curriculum development in medical education. This program at İzmir Democracy University (IDU) aims to enhance medical students' knowledge and practical skills in disaster medicine
through comprehensive theoretical and practical courses. The methodology of disaster content education varies from didactic approaches to field exercises, depending on the content and local context.

*Scott, Carson & Greenwell (2010)* study targeted fourth-year medical students with a three-hour training module that included case-based teaching, hazardous materials scenes, and sudden mass casualty events. Using a pre-test/post-test design, the study measured students' perceptions and learning degrees on disaster medicine topics. Performance evaluations involved scenarios with hazardous materials and mass casualty incidents. Students' overall knowledge levels increased from 3.76/10 pre-training to 7.64/10 post-training. Post-test scores increased by 48%, and students accurately tagged 94% of victims in a mass casualty event. The program demonstrated that students could quickly learn and apply disaster medicine topics, suggesting it as an easily integrated training program into university curricula. This study also targeted fifth-year medical students over two weeks, including theoretical and practical lessons. Using experimental and control groups, the program assessed students' disaster literacy pre- and post-training. The program improved students' knowledge and skills in disaster medicine, contributing to more responsible decision-making. It is seen as a significant intervention to enhance the ability to provide effective health services during disasters. *Scott, Carson & Greenwell (2010)* study and our study both emphasize the importance of disaster medicine education by demonstrating the effectiveness of distinct methods in improving medical students' preparedness and response capabilities during disasters. *Scott, Carson & Greenwell (2010)*'s study involved a three-hour intensive training module for fourth-year medical students, while our study utilized a two-week program combining theoretical and practical lessons for fifth-year medical students. Both highlight the critical role of education in enhancing students' disaster response knowledge and skills.

*Scott, Carson & Greenwell (2010)*'s approach, focusing on short-term, intensive training, effectively equipped students with essential disaster response capabilities in a limited time frame. In contrast, our approach, with its longer and more comprehensive program, provided deeper learning and broader skill development. These findings underline that both intensive and comprehensive methodologies can be adapted to integrate disaster medicine education into medical curricula, catering to varying institutional needs and resources. *Tas, Cakir & Kadioglu (2020)*'s study in Türkiye examined the preparedness levels of nurses for disasters and the factors influencing these levels. Conducted with 230 volunteer nurses at a public hospital in Kahramanmaraş, the study used an interview form to determine nurses' personal and professional characteristics and the Nurses' Disaster Preparedness Perception Scale. Analysis found no significant difference between nurses' gender, years of service, and disaster experience and their preparedness perception, while educational status, disaster education, and reading the hospital disaster plan showed significant differences. The study determined that nurses were partially prepared for disasters and that higher education levels, participation in disaster training programs, and reading hospital disaster plans increased preparedness levels (*Tas, Cakir & Kadioglu, 2020*).

In another study by *Wang et al. (2020)* in China, the validity and reliability of questionnaires developed for disaster nursing education were tested. This study highlighted

that disaster nursing education is crucial in meeting students' learning needs and can guide the development of these educational programs. Disaster education plays a significant role in enhancing students' effective response capabilities in disaster situations.

*Labrague et al. (2021)* study in Oman aimed to examine the factors affecting nurses' self-efficacy in disaster response. The study included 444 nurses working in specific hospitals, evaluating the impact of disaster knowledge, skills, and demographic characteristics on disaster response self-efficacy. Findings indicated that disaster education and previous experiences played a crucial role in enhancing nurses' disaster response self-efficacy. Preparing nurses for disasters and providing disaster response training enables them to intervene more effectively and safely during disasters. Therefore, widespread implementation of disaster education programs and development of nurses' disaster response skills are of great importance.

*Kozyel et al. (2018)* study examines the state and characteristics of university-level disaster management education programs in Türkiye. The study highlights the importance of disaster management education and evaluates associate, undergraduate, master's, and doctoral-level disaster education programs offered by universities. It identifies 33 disaster education programs in Türkiye, 24.2% of which provide undergraduate-level education. However, only 4.5% of these programs offer multidisciplinary disaster education and training, underscoring the limited integration of diverse disciplines. Most programs (70.3%) adopt competency-based curricula, focusing on disciplines such as hazard and risk reduction, research methods, logistics, ethics, and public health. Disaster medicine, on the other hand, is mentioned as a field primarily offered at the graduate and doctoral levels (*e.g.*, the "Disaster Medicine Doctoral Program" at Bezm-i Alem University). Undergraduate programs, however, fall under broader "Emergency Aid and Disaster Management" categories, encompassing a more general disaster management education.

Our study addresses these gaps by providing a truly multidisciplinary model that integrates a wide array of disciplines. The program we developed includes forensic medicine, public health, pediatrics, general surgery, obstetrics and gynecology, radiology, nutrition and dietetics, physical therapy and rehabilitation, psychiatry, and child and adolescent mental health. Additionally, it involves diverse professional groups such as nurses, paramedics, AFAD managers, Red Crescent managers, and firefighters. Supporting more than 20 disciplines, this program sets a new standard in disaster medicine education, offering a robust and innovative approach that emphasizes interdisciplinary collaboration. This breadth and integration make our model significantly more comprehensive and impactful, addressing the critical need for coordination and effective training across various fields in disaster management.

All these studies emphasize the importance of disaster education and show that educational programs are effective in preparing health professionals, not only medical students but also other healthcare workers, for disasters. Disaster medicine education for both medical students and nurses is critical in developing effective response capabilities in disaster situations. Regular education programs and drills will ensure that both students and healthcare professionals are better prepared for disasters (*Kozyel et al., 2018*; *Wang et al., 2020*; *Tas, Cakir & Kadioglu, 2020*; *Labrague et al., 2021*). These studies highlight the

necessity of integrating disaster medicine education into curricula. Future research should examine the long-term effects of such educational programs and their impact on different student groups. Additionally, considering cultural differences and regional needs will make educational programs more effective.

## CONCLUSIONS

The world has become a place where any disaster can occur anywhere and anytime. In schools of medicine in countries at risk of disasters such as Türkiye, a disaster medicine training program that is ultimately officialized as an elective course in the curriculum can educate medical students about disaster medicine. It is also important to plan studies on disaster literacy for physicians, who are in the team of first responders. Increasing the disaster literacy levels of physicians contributes to a decrease in disaster-associated risks by increasing the knowledge and skills of society regarding disasters. In our study, the disaster literacy levels of the participants in the experimental group were found to increase significantly because of the training program on disaster medicine. These results showed that the program that was organized was effective and played a significant role in improving disaster literacy skills.

## ACKNOWLEDGEMENTS

The initial findings of this study were presented as an oral presentation at the 5th International Medical Congress of Izmir Democracy University, held in Izmir, Turkey, on December 1–3, 2023.

This article was conducted as part of the Disaster Medicine Internship at Izmir Democracy University Faculty of Medicine. We extend our deepest gratitude to all the medical students who contributed to this study and demonstrated their dedicated participation at every stage of the research. Additionally, we express our sincere thanks to the university administrators and faculty members who supported us throughout the research process and provided the necessary resources. The translation of this manuscript from Turkish to English involved the use of an AI-based language model (ChatGPT). The authors reviewed and edited the content to ensure accuracy and fluency during this process.

### Funding

The authors received no funding for this work.

### Competing Interests

The authors declare there are no competing interests.

### Author Contributions

- Aysel Başer conceived and designed the experiments, performed the experiments, analyzed the data, prepared figures and/or tables, authored or reviewed drafts of the article, and approved the final draft.

- Zeynep Sofuoğlu conceived and designed the experiments, performed the experiments, authored or reviewed drafts of the article, english translate, and approved the final draft.

## Human Ethics

The following information was supplied relating to ethical approvals (i.e., approving body and any reference numbers):

The data were collected from Izmir University of Medicine students after obtaining informed consent. The study was conducted in compliance with the ethical standards set forth by the İzmir Democracy University Ethics Committee. Ethical approval was secured prior to the initiation of the study (Ethics Committee Approval No. 2023-02-133, Date: 22.02.2023).

## Data Availability

The raw data is available in the Supplementary Files.

## Supplemental Information

Supplemental information for this article can be found online at http://dx.doi.org/10.7717/peerj.18800#supplemental-information.

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
