# Peer review of "The impact of a disaster medicine clinical training program on medical students’ disaster literacy"

_PeerJ, doi:10.7717/peerj.18800_

## Round 0.1 · original submission · Major Revisions

The manuscript addresses a topic of international relevance by evaluating the effectiveness of a disaster medical education program using a literacy scale. According to the reviewers, the text is clearly written, provides sufficient context on the topic with adequate references, and makes the dataset available in an accessible and well-explained format. However, several limitations were identified that must be addressed to meet the expected quality standards.

Reviewer 3 highlighted that, although the study fits well within the journal's scope and the research question is relevant, the clarity of the study design and analytical approach needs improvement. They pointed out that the comparison between the experimental and control groups, conducted through paired and independent t-tests, lacks a quantitative baseline verification of the groups at the pre-test stage. This verification is crucial to avoid biased conclusions regarding the effectiveness of the intervention. Additionally, the reviewer suggested that the authors consider using a "difference-in-differences" approach to strengthen the analysis and enhance the validity of the inferences.

Reviewer 4 raised important concerns related to the methodological description. They noted that the manuscript does not clarify the criteria for selecting participants for the experimental and control groups, the number of interventions conducted, the content of the educational sessions, or the follow-up procedures used. Furthermore, the reviewer flagged the possibility of bias due to interaction between students from the experimental and control groups, as they belonged to the same institution, which might have influenced post-test results. The reviewer also pointed out inconsistencies in the text's organization, such as the tables being cited out of order, and criticized the lack of references for several claims in the discussion section. They further suggested that titles of references in languages other than English should be translated or presented in bilingual format.

Both reviewers emphasized the need for greater methodological detail and analytical adjustments to improve the validity of the findings. They also agreed that the absence of citations for studies mentioned in the discussion weakens the argument and requires immediate attention.

In light of the reviewers' contributions, it is recommended that the authors undertake a thorough revision of the manuscript, with particular focus on the following points:

Provide a detailed description of the study design, including participant selection criteria, the content of the interventions, and strategies to control potential biases.
Reanalyze the data using a more robust approach, such as a difference-in-differences method, to assess the effectiveness of the intervention.
Revise the text organization, ensuring tables are cited in the correct order and maintaining consistency in the narrative.
The manuscript has merit and addresses a relevant topic, but substantial revisions are necessary to meet the journal's standards. My initial inclination, considering the methodological weaknesses, is to recommend rejection of the manuscript in its current form. However, I am hopeful that the authors will be able to adequately address these issues, and I encourage them to revise the work thoroughly and resubmit, as the study holds significant potential.

·

Basic reporting

It was clear. the references were appropriate. It was well written.

Experimental design

This was a nice case control design with clear objectives.

Validity of the findings

The findings appear valid.

Additional comments

Nice paper with good recommendations that show the utility of disaster training

This is a nice and focused paper with a clear purpose. The conclusions are reasonable. I have no criticism and suggestions for modification.

·

Basic reporting

Dear Authors,
I read your work with interest. Today, with the increase in disasters, various health problems arise. Therefore, there is a great need for health professionals. I find the implementation of a Disaster Medicine internship especially for physicians successful and your findings show that the desired goal has been achieved. The study can be strengthened a little more with a few suggestions.
1. When postgraduate education in Turkey is investigated, it is seen that there are Disaster Medicine master's and doctoral programmes. Could you compare these programmes with your educational content.
2. There is a missing word sentence on page 12 line 289.
3. I am aware that you have translated the DLS scale into the appendices. Could you kindly clarify whether you obtained this scale from the relevant article's appendices or from the scale's creator? It is essential to access the original English version of the scale.

Experimental design

no comment

Validity of the findings

no comment

Additional comments

no comment

Reviewer 3 ·

Basic reporting

The present study quantitatively evaluates the effectiveness of a disaster medical education program using a literacy scale. This is a topic of international significance.

The manuscript is written in clear English, provides sufficient background of the topic with adequate references, and indicates the important results for disaster medicine education and literacy. The dataset has been made available in an appropriate format, with accompanying explanations in English. The authors have submitted their manuscript in appropriate manner.

On the other hands, the study design and analytical framework are difficult to follow. Particularly regarding the design, a fundamental improvement is necessary.

Experimental design

The present study aligns with the fields of health sciences or medicine and, therefore, fits within the Aims and Scope of this journal. The research question, which examines the effectiveness of disaster medical education from a literacy perspective, seems appropriately framed, and no ethical concerns are immediately apparent. However, as previously noted, certain technical issues appear within the study’s research and analytical design.

In this study, a comparison is made between an experimental group that underwent disaster medical training and a control group that did not receive such training. With training as the independent variable and literacy improvement as the outcome, this comparative approach forms the core of the study’s design. The authors conduct two primary analyses: first, they use a paired t-test to assess pre-post literacy changes within the experimental group; second, they apply an independent samples t-test to compare post-training literacy levels between the experimental and control groups. Based on these results, the authors assert the effectiveness of the training; however, this assertion may need to be clearer regarding the literacy baseline in the control group at the pre-test stage.

Please consider the following hypothetical scenario: at the pre-test stage, the experimental group’s literacy score is 10, the control group’s score is 1; at the post-test stage, the experimental group’s score is 12, while the control group’s score is 8. In this case, despite not receiving training, the control group’s literacy improvement exceeds that of the experimental group. Furthermore, even if there is a statistically significant difference in post-test literacy scores between the experimental and control groups, such a difference could be attributed to initial disparities between those groups rather than training efficacy. In light of this, it may be scientifically challenging to attribute observed outcomes solely to the training effect.

This design limitation, as noted, may not be sufficiently addressed by the authors’ statement that there are no differences in knowledge or skill levels between the control (3rd- and 4th-grade students) and experimental (5th-grade students) groups. Instead of a descriptive claim, the need for quantitative verification becomes increasingly apparent. This is methodological and scientific weak point of the present study.

I believe that this paper's research question is excellent. Reanalyzing the data and resubmitting the manuscript could be an option. By confirming that there is no difference in literacy levels between the two groups at the pre-test stage and then comparing the literacy levels of both groups at the post-test stage or by comparing the change in literacy levels from pre- to post-test between the experimental and control groups (i.e., a ‘difference-in-differences’ analysis), this study could be well-positioned for future publication.

Validity of the findings

The introduction is persuasive, and a discussion ties the findings to the results. However, as mentioned previously, interpreting the results directly remains challenging.

Additional comments

Tables 1 and 2 appear to be presented in reverse order; Table 2 is referenced earlier in the text.

·

Basic reporting

Overall impression
If you talk about disaster, keep the specific example of the disaster because different disaster events have different nature, and hazard profiles.

General comments
• I think the previous name of Turkey has been changed to Turkiye. Keep Methods in past tense as the research is already complete. Many statements in the Discussion section need citation to support these.

Experimental design

Methods
• It is not clear how you selected participants for experimental and control groups. In other words, mention sampling techniques adopted for both groups of participants. What were the contents in the intervention session? How many interventions did you execute among the same group during the study period? How many follow-ups did you do? How did you maintain integrity of the control groups during the whole procedure? What was the ratio of control: experimental group? Which disasters did you cover during theoretical and practical demo education sessions?
• You selected the 5th-year medical students in the experimental group and 3rd- and 4th-year medical students in the control group. Since all students were from the same university, there might be chances that the students in both groups interacted with each other about the research intervention. This might have affected the post-test impact of the intervention. How did you nullify this probability of getting biased results with the possible interaction of the students among different groups?
• Lines 125-127: You have mentioned that the theoretical sessions were 38 hours long and practical sessions were 20 hours long. This means, total 58 hours (theory and practical) intervention sessions were conducted among the experimental group? Were the control group also exposed to the theoretical course or not?
• Lines 161-162: You have cited Table 2 before Table 1.

Validity of the findings

no comment

Additional comments

Discussion
• Line 250: You have mentioned “There are several studies on the exposure of students to various disasters” but have not cited references to those studies.
• Lines 250-253: Clarify the disaster event, the year it happened and hazards it caused.
• Lines 256-257: You have mentioned “… the rates of exposure to disasters were much lower than those reported in other studies.” What are those ‘other studies’? What are the frequencies of disasters in those studies? Which disasters did they mention?
• Lines 263-264: You have mentioned “Other studies have examined the disaster preparedness and disaster awareness levels of university students” but have not cited references to those ‘other studies’.
• Lines 264-266: How did you confirm the statement “In general, students have been determined to have high levels of disaster awareness and low levels of disaster preparedness”?
• Lines 270-271: You have motioned “During the Anesthesia and Reanimation Conference in May 2013 in Copenhagen, Denmark, a 3-day disaster medicine course was offered to students.” Did you offer the course yourself? If not, cite the reference.
• Lines 299-301: You have cited ‘Lancer et al.'s 2010 study’ but have not kept its details in the References section. This also applies in lines 314-315.
• Lines 312-313: What are those ‘both studies’?
• Lines 315-316: What are those ‘both approaches’?
• Lines 331-332: You have cited ‘Leodoro et al.'s study in Oman’ but have not kept its details in the References section.

References
• References 6, 10, 12, 15-17: Please replace the references with English reference, if possible. If not, at least keep the title also in English. Keep date of access (in reference 10). The authors, editors and the reviewers may feel difficulty to cross-verify the references if these are not in English language.

---

## Round 0.2 · accepted · Accept

Thank you for submitting the revised version of your manuscript. After carefully reviewing the changes, I can confirm that all reviewers' comments and suggestions have been appropriately addressed.

Based on this assessment, I am pleased to inform you that the manuscript is now ready for publication. Congratulations.

·

Basic reporting

The study is appropriate.

Experimental design

The study is appropriate.

Validity of the findings

The study is appropriate.

Additional comments

None.

·

Basic reporting

no comment

Experimental design

no comment

Validity of the findings

no comment

Additional comments

The authors have satisfactorily addressed all the concerns that I raised in my previous comments. Now, the revised manuscript seems much improved. Hence, it can be accepted.